# A Bibliometric Analysis of the Trends and Characteristics of Railway Research

**Diogo Da Fonseca-Soares** [1,2,3,*] , **Josicleda Domiciano Galvinicio** [2], **Sayonara Andrade Eliziário** [4] **and Angel Fermin Ramos-Ridao** [3]

1   Brazilian Company of Urban Trains, Square Napoleao Laureano, 01, Varadouro, João Pessoa 58010-150, Brazil
2   Department of Geographical Sciences, Federal University of Pernambuco, Avenue Professor Moraes Rego, 1235, Cidade Universitária, Recife 50670-901, Brazil
3   Civil Engineering Department, University of Granada, Campus de Fuentenueva, s/n. E.T.S Ingenieros de Caminos, Canales y Puertos, 18071 Granada, Spain
4   Department of Renewable Energy Engineering, Federal University of Paraíba, Campus I, Castelo Branco III, João Pessoa 58051-900, Brazil
*   Correspondence: diogosoares@cbtu.gov.br; Tel.: +55-83-98701-6734

**Abstract:** A retrospective bibliometric analysis of the railway sector covering the 20-year period between 2002 and 2021 was carried out to better understand the characteristics of the railway research. The Scopus database contained 1918 articles published with the keywords "Rail System". VOSviewer software was used to create network maps from each of the variables studied. The results showed a huge increase in the number of publications over this period—notably, work written by Zhang, Y.T., who was found to be the most productive author. Engineering was found to be the most studied subject area of knowledge; *Transportation Research Record* was the journal with the highest number of publications; and China was revealed to be the leading country regarding this research field, Southwest Jiaotong University being the leading institution in this topic. Finally, there was a lack of research on the environmental impact and sustainability of railway systems, an area that could be opened up for future study.

**Keywords:** rail system; literature review; transportation sector; sustainability and development

## 1. Introduction

The railway system is the most efficient mode of human transportation since it serves as a mass transportation system for the general population and reduces traffic congestion in major cities. It is also useful for cargo logistics, allowing for the transport of a variety of cargo in large quantities and with minimal delay [1].

Rail systems are a fundamental economic component of most countries, as they can transport millions of passengers and millions of dollars in goods from their origins to destinations every day. According to many empirical works and articles, rail transport, which produces very low $CO_2$ emissions, is one of the most environmentally friendly and safest modes of transport [2–12].

The expansion of rail networks is an important issue for the economic and social development of any country, as it enables revenue and jobs to be created sustainably and saves energy [11,13,14].

People's commuting time may be spent contributing to society or doing something that would improve their quality of life. According to several studies, the time saved on journeys to work is commonly invested in a revenue-generating activity [15,16]. In this respect, the railway system improves people's capacity for movement, which is a fundamental right, while also providing more time for the population to dedicate to work or personal pursuits [17].

Human mobility is an incredibly important issue, and as railway systems are a form of mass transport, many studies have been undertaken and many types of technology have been generated to improve the public image of these systems. Nevertheless, the popularity of public rail transport is still low, despite the huge investments that it has received in recent decades [18]. Therefore, studying this system is extremely important to promoting greater public acceptance. If the popularity of rail transport increased, the problems related to traffic could be reduced. This would have a positive environmental impact, as buses, trucks, and cars significantly increase the emissions of greenhouse gases. Rail travel produces much lower levels of emissions, and its increased use could lead to urban improvement [19].

Different factors associated with rail travel need to be improved and studied to improve railway systems and their relationship with urban planning. Some of these factors are: travel costs [20], quality of service [21], punctuality, number of connections, distances from stations to home or work, sustainability, energy efficiency [22], renewable energy use [23], and high-speed trains [24].

Therefore, a bibliometric analysis of the railway system sector is important to better understand what is being investigated and what gaps in the research exist so that they can be studied in the future. Current bibliometric research on railways exists that focuses on different areas. Some of these are: the socioeconomic impact of transport [25]; logistics, in order to explore the current research on sustainable logistics technology for decision making in organizations [26]; monitoring analysis of the state of the rail transport system, in order to present updated content of the sensors used in this monitoring [1]; and worker–railway relationships, in order to explore the research using the human reliability tool (HRA) applied to the rail system [27].

This study aims to examine the development of the field of bibliometrics by considering a 20-year period of global railway research to better understand the research that exists on railway systems. In order to outline the bibliometric research on railway systems and future research trends, the parameters were set by using the questions:

Q1. What publication trends exist regarding rail systems?

Q2. What journals published the most articles about rail systems?

Q3. Who contributes the most (authors, institutions, and countries) to the study of railway systems?

Q4. What are the main research areas on rail systems?

## 2. Materials and Methods

The methodology used in this study is bibliometric analysis, which is a technique for assessing the output of publications in a particular field of knowledge. It maps academic communities and discovers networks of researchers and their different motivations [28].

The bibliometric analysis can be achieved by developing indicators that summarize the most prolific institutions and authors, the most cited academics, and co-authorship networks regarding publications. The extraction of metadata to evaluate the progress of a field of knowledge over time is another method that is used. [28–31]

The articles search for the bibliometric analysis took place in January 2022 and was divided into four stages, as shown in Figure 1.

Selection criteria: At this stage, the first criteria for the bibliometric search were defined: articles in the research field of "rail system" from the last 20 years (from 2002 to 2021).

Choice of Database: there are numerous types of databases for academic documents; however, Scopus was chosen for this study because it is a scientific database containing a large number of publications, authors, and journals that meet peer-reviewed scientific quality standards. [32–36].

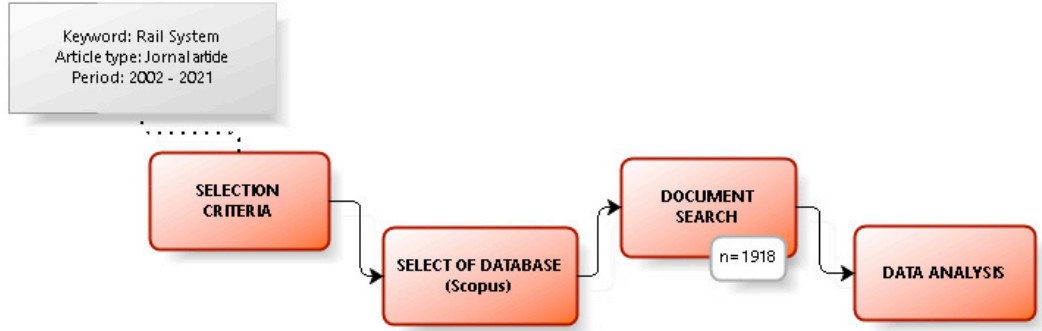

**Figure 1.** Stages of the articles search for the bibliometric analysis.

Document search: The first search for "Rail System" resulted in 4510 documents for the 20-year timeframe. Following this search, other filters were applied, and 1918 documents met the criteria, as shown in Table 1. The final enter query string was:

**Table 1.** Summary of searches performed on the Scopus database.

| Searches | Keyword | Filter | Result | Comment |
|---|---|---|---|---|
| 1st | Rail System | | 4510 | Not very discriminatory (Extremely high number of documents) |
| 2nd | Rail System | Year: Between 2002 and 2021 | 3513 | A 20-year dataframe was chosen, as indicated by most authors for bibliometric analysis in this field of science |
| 3rd | Rail System | Document type: Articles | 1943 | Articles pass through peer-review, which indicates higher quality |
| 4th | Rail System | Publication stage: Final | 1918 | Documents under publishing processes were not included. |

TITLE-ABS-KEY ("rail system") AND (LIMIT-TO (PUBYEAR, 2021) OR LIMIT-TO (PUBYEAR, 2020) OR LIMIT-TO (PUBYEAR, 2019) OR LIMIT-TO (PUBYEAR, 2018) OR LIMIT-TO (PUBYEAR, 2017) OR LIMIT-TO (PUBYEAR, 2016) OR LIMIT-TO (PUBYEAR, 2015) OR LIMIT-TO (PUBYEAR, 2014) OR LIMIT-TO (PUBYEAR, 2013) OR LIMIT-TO (PUBYEAR, 2012) OR LIMIT-TO (PUBYEAR, 2011) OR LIMIT-TO (PUBYEAR, 2010) OR LIMIT-TO (PUBYEAR, 2009) OR LIMIT-TO (PUBYEAR, 2008) OR LIMIT-TO (PUBYEAR, 2007) OR LIMIT-TO (PUBYEAR, 2006) OR LIMIT-TO (PUBYEAR, 2005) OR LIMIT-TO (PUBYEAR, 2004) OR LIMIT-TO (PUBYEAR, 2003) OR LIMIT-TO (PUBYEAR, 2002)) AND (LIMIT-TO (DOCTYPE, "ar")) AND (LIMIT-TO (PUBSTAGE, "final")).

Data analysis: To develop the bibliometric analysis, the data obtained from the Scopus database were examined. To determine the level of interest in the research area, the year of publishing was analyzed, along with the journal, the area, the author and co-authors, the institution, the country, the keywords used in the research, the citation count, the H-index, and the article influence score from the Scimago Journal Rank (SJR).

Finally, the VOSwiever program was used to create network maps for each parameter, enabling word classification and analysis [1,25,27,37,38].

## 3. Results and Discussion

### 3.1. What Publication Trends Exist Regarding Rail Systems (Q1)?

This section summarizes the key characteristics of scientific production about "Rail system" in terms of the evaluation of the total number of articles (A), authors (AU), countries (C), citations (TC), average citations per article (TC/A), and journals (J). The analysis covered a 20-year period, from 2002 to 2021, which was separated into 5-year periods to facilitate the analysis.

As a result, Table 2 presents the evolution of these articles' main characteristics. If we focus on the total number of articles published (A), it increased by about 56% in the second five-year period compared to the first and by approximately 50% in the third five-year period compared to the second. In relation to the previous five-year period, the most recent five-year period increased by 20%.

**Table 2.** Characteristics of the scientific literature on railway systems.

| Year | Articles (A) | Authors (AU) | Countries (C) | Citations (TC) | TC/A | Journals (J) |
|---|---|---|---|---|---|---|
| **2002–2006** | 248 | 550 | 41 | 212 | 0.85 | 149 |
| **2007–2011** | 387 | 901 | 72 | 1535 | 3.97 | 242 |
| **2012–2016** | 583 | 1570 | 57 | 5824 | 9.99 | 325 |
| **2017–2021** | 700 | 1992 | 62 | 15,334 | 21.91 | 381 |

(TC/A): Average citations per article.

Overall, there is a 126% increase in the total number of articles published from the first five years of the study (2002–2006) to the last five years (2017–2021). Figure 2 illustrates the exponential growth in the number of articles published during the previous 20 years.

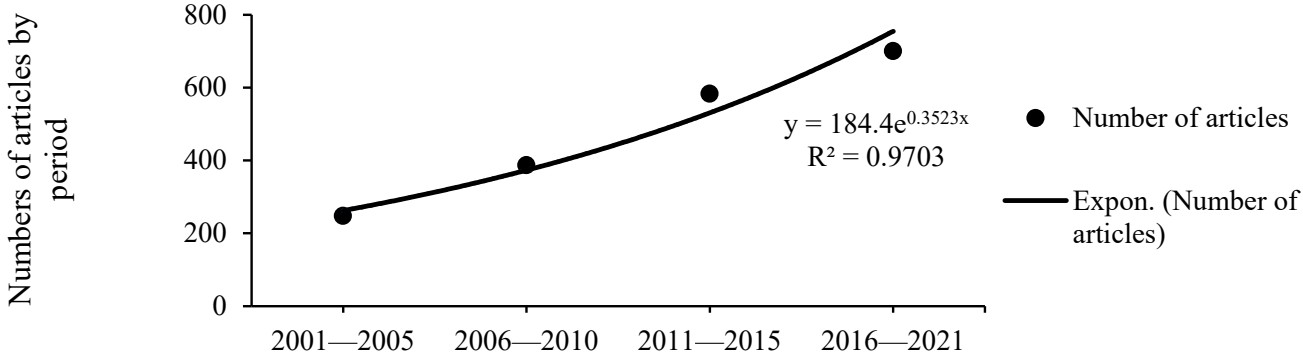

**Figure 2.** Trends of articles published on rail systems over the last two decades.

All other variables have likewise increased: authors (A), countries (C), citations (TC), citations per article (TC/A), and journals (J). The number of countries paying attention to the issue grew by nearly 50% in 20 years, from 41 to 62. The number of citations increased by 624% between 2002–2006 and 2007–2011, indicating a significant rise in interest in the research network during this time. The rise from 2007–2011 to 2012–2016 was 279%, while the increase from 2012–2016 to 2017–2021 was 163%.

The most significant increase in the number of journals (J) occurred from 2002–2006 to 2007–2011 (62%), followed by 34% from 2007–2011 to 2012–2016 and 17% from 2012–2016 to 2017–2021.

As a result, the indicators analysis indicates that there is a global interest in increasing research production in the topic.

Between 2002 and 2021, 27 areas of knowledge were researched, and Figure 3 shows the five main thematic areas that Scopus included in these articles. Throughout the study period, the Engineering category received the highest research production, with a total of 1.243, accounting for 36.43% of the total, followed by Social Science (529, 15.50%), Environmental Science (222, 6.50%), Informatics (172, 5.04%), and Materials Science (165, 4.83%). Furthermore, studies on rail system have been published in a variety of fields, reflecting the vast span of the research.

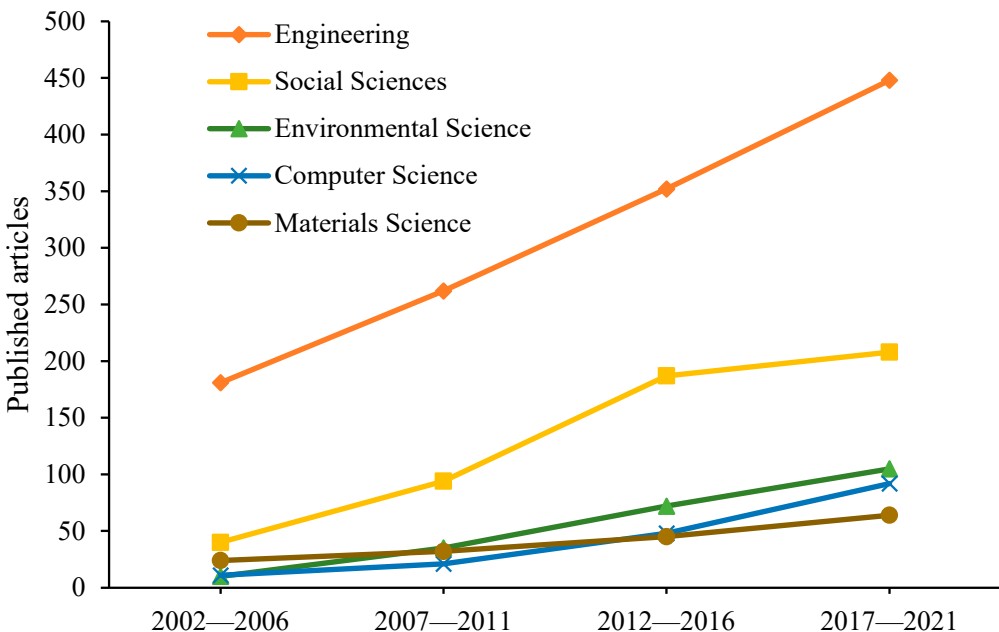

**Figure 3.** Evolution of the number of articles published on railway systems by subject area.

*3.2. What Journals Published the Most Articles about Rail Systems? (Q2)*

For this section, the 20 journals that published the most on rail systems were selected. The key characteristics of the articles published in these scientific journals are shown in Table 3. In the year 2020, 40% of them were in the first quartile (Q1) of the SJR index. These Q1 journals published 170 articles, accounting for 36% of all the articles published in this field in the top 20 journals. Moreover, journals in the second quartile (Q2) published 35% of all the articles published. The journals in the fourth quartile (Q4) accounted for 25% of the publications.

When we analyzed the impact factor, we observed that *Fuel* from the Netherlands has the highest H-index among journals (HJ), followed by *Urban Studies* from the United Kingdom.

The journals that published the largest number articles were *Transportation Research Record*, *Neiranji Gongcheng Chinese Internal Combustion Engine Engineering*, and the *Journal of Transport Geography*, with 70 (15.09%), 51 (10.99%), and 39 (8.41%) articles published, respectively.

Around 55% of the journals showed continuity in their publication rates, with at least one article published every five years, which shows that railway systems were well studied during this period.

In the last five years, *Sustainability* (Switzerland) published the highest number of articles on rail systems. Because the journal's specialty is Environmental Sciences, this shows a tendency for studies on rail systems and sustainability challenges. This might be tied to the growing concern about the effects of global climate change and the shortage of natural resources, both of which are causing increasing limits on economic activity, implying the need for continuous improvement in services and the means of production [39].

Finally, it is worth noting that the largest number of journals come from the United Kingdom (40%), followed by China and the United States.

**Table 3.** Key features of articles published in the 20 journals with the highest scientific production.

| Journals (J) | A | TC | TC/A | H | HJ | SJR | C | First Article | Last Article | Articles by Five-Year Period | | | |
|---|---|---|---|---|---|---|---|---|---|---|---|---|---|
| | | | | | | | | | | 2002–2006 | 2007–2011 | 2012–2016 | 2017–2021 |
| *Transportation Research Record* | 70 | 608 | 8.69 | 15 | 119 | 0.62 (Q2) | United States | 2015 | 2020 | 9 | 28 | 22 | 11 |
| *Neiranji Gongcheng Chinese Internal Combustion Engine Engineering* | 51 | 114 | 2.24 | 5 | 14 | 0.18(Q4) | China | 2013 | 2021 | 10 | 13 | 20 | 8 |
| *Journal Of Transport Geography* | 39 | 1071 | 27.46 | 19 | 108 | 1.81 (Q1) | United Kingdom | 2016 | 2021 | 3 | 6 | 17 | 13 |
| *Transportation Research Part A Policy And Practice* | 30 | 1471 | 49.03 | 16 | 133 | 2.18 (Q1) | United Kingdom | 2014 | 2021 | 2 | 6 | 10 | 12 |
| *Tiedao Xuebao Journal Of The China Railway Society* | 27 | 152 | 5.63 | 7 | 30 | 0.40 (Q2) | China | 2015 | 2021 | 0 | 4 | 11 | 12 |
| *Sustainability Switzerland* | 25 | 203 | 8.12 | 9 | 85 | 0.61 (Q1) | Switzerland | 2018 | 2021 | 0 | 2 | 2 | 23 |
| *Proceedings Of The Institution Of Mechanical Engineers Part F Journal Of Rail And Rapid Transit* | 23 | 327 | 14.22 | 8 | 55 | 0.66 (Q2) | United Kingdom | 2009 | 2021 | 1 | 3 | 4 | 15 |
| *Zhongguo Tiedao Kexue China Railway Science* | 21 | 140 | 6.67 | 8 | 27 | 0.44 (Q2) | China | 2004 | 2021 | 7 | 4 | 2 | 8 |
| *Eb Elektrische Bahnen* | 18 | 30 | 1.67 | 3 | 12 | 0.10 (Q4) | Germany | 2003 | 2018 | 10 | 3 | 3 | 2 |
| *Neiranji Xuebao Transactions Of CSICE Chinese Society For Internal Combustion Engines* | 18 | 51 | 2.83 | 4 | 18 | 0.16 (Q4) | China | 2002 | 2020 | 6 | 3 | 5 | 4 |
| *Transport Policy* | 18 | 478 | 26.56 | 11 | 96 | 1.69 (Q1) | United Kingdom | 2007 | 2021 | 0 | 2 | 8 | 8 |
| *Fuel* | 16 | 580 | 36.25 | 10 | 213 | 1.56 (Q1) | Netherlands | 2005 | 2021 | 0 | 1 | 7 | 8 |

**Table 3.** *Cont.*

| Journals (J) | A | TC | TC/A | H | HJ | SJR | C | First Article | Last Article | Articles by Five-Year Period | | | |
|---|---|---|---|---|---|---|---|---|---|---|---|---|---|
| | | | | | | | | | | 2002–2006 | 2007–2011 | 2012–2016 | 2017–2021 |
| *Research In Transportation Economics* | 14 | 128 | 9.14 | 6 | 46 | 1.02 (Q1) | United States | 2007 | 2021 | 0 | 0 | 9 | 5 |
| *Transportation Research Part D Transport And Environment* | 14 | 202 | 14.43 | 8 | 99 | 1.60 (Q1) | United Kingdom | 2011 | 2021 | 0 | 2 | 5 | 7 |
| *Urban Studies* | 14 | 677 | 48.36 | 10 | 147 | 1.92 (Q1) | United Kingdom | 2005 | 2020 | 1 | 2 | 4 | 7 |
| *Planning* | 13 | 7 | 0.54 | 1 | 11 | 0.10 (Q4) | United States | 2004 | 2018 | 1 | 6 | 4 | 2 |
| *Transportation Planning And Technology* | 13 | 155 | 11.92 | 7 | 42 | 0.43 (Q2) | United Kingdom | 2004 | 2020 | 1 | 5 | 3 | 4 |
| *Urban Rail Transit* | 13 | 126 | 9.69 | 5 | 14 | 0.52 (Q2) | Germany | 2015 | 2021 | 0 | 0 | 3 | 10 |
| *ZEV Rail Glasers Annalen* | 13 | 4 | 0.31 | 1 | 6 | 0 | Germany | 2002 | 2008 | 8 | 5 | 0 | 0 |
| *Energies* | 12 | 49 | 4.08 | 3 | 93 | 1.60 (Q2) | Switzerland | 2015 | 2021 | 0 | 0 | 1 | 11 |

(J): journals; (A): number of articles; (TC): number of citations; (TC/A): average number of citations per article; (H): H-index of articles; (HJ): H-index of journals; (SJR): Scimago Journal Rank (quartile); (C): country.

### 3.3. Who Contributes the Most (Authors, Institutions, and Countries) to the Study of Rail Systems? (Q3)

This subsection highlights the most productive individual authors, as well as their co-operation with their peers, focusing on co-authorship indicators. These indicators highlight the most productive institutions and nations, the co-authorship network, and the most successful international alliances.

Some of most prolific authors on the rail system between 2002 and 2021 are shown in Table 4. The author with the most works to his name is Zhang, Y.T., who published 15 articles and was cited 45 times during this period; he is followed by Ma, X., with 14 articles and 55 citations. Both authors share the same research quality indicator (H-index = 4), and Wang, P., who published 13 articles with 59 citations, has a research quality indicator of H-index = 5. Pagliara, F. has fewer publications (9) to his name than the author in the highest position but has a research quality indicator of H-index = 5.

**Table 4.** The most prolific authors writing about rail systems between 2002 and 2021.

| Autores | A | TC | TC/A | Institution | C | 1st A | Last A | H-Index |
|---|---|---|---|---|---|---|---|---|
| Zhang, Y.T. | 15 | 45 | 3.00 | Southwest Jiaotong University | China | 2005 | 2018 | 4 |
| Ma, X. | 14 | 55 | 3.93 | Beijing Jiaotong University | China | 2013 | 2021 | 4 |
| Wang, P. | 13 | 59 | 4.54 | Ministry of Education China | China | 2013 | 2020 | 5 |
| Ouyang, G.Y. | 12 | 35 | 2.92 | Beijing Institute of Technology | China | 2005 | 2013 | 4 |
| Fan, L. | 11 | 22 | 2.00 | Naval University of Engineering | China | 2013 | 2021 | 3 |
| Mulley, C. | 11 | 272 | 24.73 | Harbin Engineering University | China | 2007 | 2019 | 2 |
| Bai, Y. | 10 | 21 | 2.10 | Shanghai Jiao Tong University | China | 2013 | 2021 | 3 |
| Chen, G.X. | 9 | 96 | 10.67 | Tianjin University | China | 2011 | 2021 | 5 |
| Huang, Z. | 9 | 65 | 7.22 | Tongji University | China | 2006 | 2018 | 4 |
| Pagliara, F. | 9 | 246 | 27.33 | Newcastle University | United Kingdom | 2009 | 2019 | 5 |

(A): number of articles; (TC) number of citations; (TC/A): average number of citations per article; (C): country; (1st A): first published article; (Last A): last article published; (H-index): Hirsch index for this author.

Among the top ten contributors, Pagliara, F. received the most citations per publication (TC/A = 27.33), followed by Mulley, C. (TC/A = 24.73). However, it is important to note that the two authors who were cited the most were only in the tenth and sixth positions in terms of number of articles published (A). This shows that the number of publications did not relate to a good number of citations and that the indices (TC/A and A) for the rail system field were not proportionate.

It is interesting to note that 90% of the ten most prolific authors are from Chinese Institutions, and only one author from this group is from an institution from the United Kingdom.

Around 40% of the most prolific authors (top ten) only began to be published in the third 5-year period of the 20-year research period, and only 40% of these authors were published in 2021.

Finally, Figure 4 displays the cooperation network between the key authors who had published on the rail system, based on the study of cooperation and the inclusion of at least three co-authorship publications, generating a total of 9 clusters and 91 authors. The colors symbolize the working groups, and the size of the circle varies according to the number of articles contributed by each author.

The analysis of cooperation between authors helps us to discover the structure of cooperation between authors and institutions, as well as to understand the links between researchers and how knowledge is spread, making it a useful analytical lens [37]. Partnerships can help spark new research by facilitating the exchange of ideas, and synergies can increase the number of opportunities for publication in high-quality journals [28].

Among the top ten most productive authors, Mulley, C., Chen, G.X., and Pagliara, F. appeared in any of the network clusters, indicating that there is strong international cooperation in transportation systems. However, it is noted that interactions are much more frequent between authors from the same country; for example, we have authors from China who are mostly co-authors from the same country (China). Interaction with

other international authors may promote the rapid growth of the research area and the dissemination of knowledge.

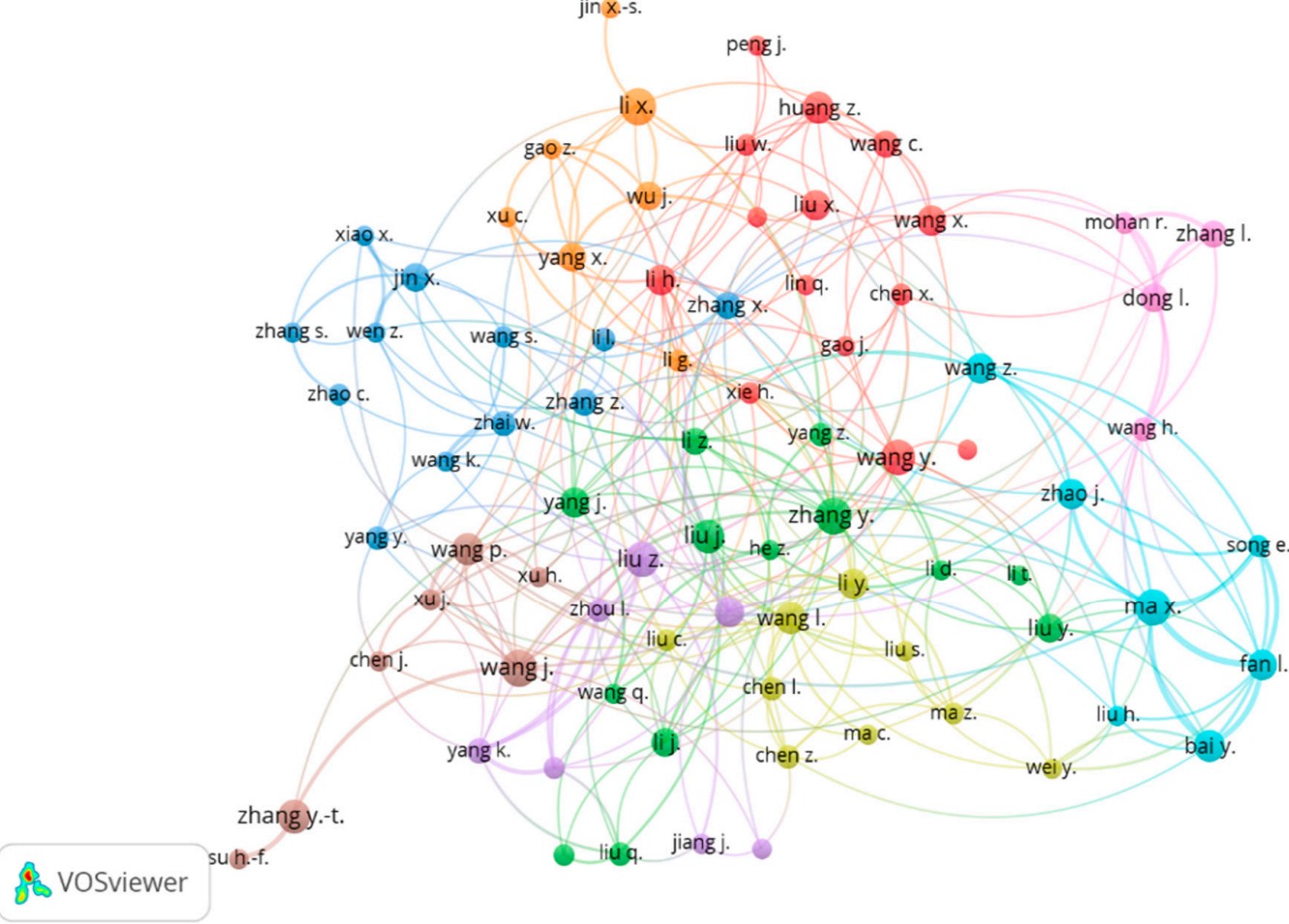

**Figure 4.** Co-authoring network of authors.

The institutions that published the most on rail systems between 2002 and 2021 are shown in Table 5. The Southwest Jiaotong University, with 106 articles and 987 citations, publishes the most on rail systems. The second on the list is Beijing Jiaotong University, with 68 articles and 568 citations, followed by the Chinese Ministry of Education, with 37 articles and 213 citations.

Newcastle University, which appears in the last position on the list with only 19 publications, has the highest average number of citations per article (37.58). This is a result of the fact that even though it had fewer publications, its articles were highly cited.

The Southwest Jiaotong University and Beijing Jiaotong University are the ones with the highest research quality rating, with H-indexes of 17 and 12, respectively. However, Newcastle University, despite its position regarding the number of publications, is in third position for research quality, with an H-index of 10.

Nine of the ten most important institutions when considering work published in this study area are in China, with only one from the United Kingdom, which indicates China's prominence in worldwide publications on rail systems.

**Table 5.** The most prolific institutions for the publications on rail systems between 2002 and 2021.

| Institution | C | A | TC | TC/A | H index | IC (%) | TC/A | |
| | | | | | | | IC | NIC |
|---|---|---|---|---|---|---|---|---|
| Southwest Jiaotong University | China | 106 | 987 | 9.31 | 17 | 12.3% | 12.62 | 8.85 |
| Beijing Jiaotong University | China | 68 | 568 | 8.35 | 12 | 22.1% | 16.87 | 5.94 |
| Ministry of Education China | China | 37 | 213 | 5.76 | 9 | 18.9% | 10.57 | 4.63 |
| Beijing Institute of Technology | China | 35 | 108 | 3.09 | 6 | 5.7% | 0.00 | 3.27 |
| Naval University of Engineering | China | 29 | 48 | 1.66 | 4 | 0.0% | 0.00 | 1.66 |
| Harbin Engineering University | China | 26 | 75 | 2.88 | 5 | 26.9% | 5.57 | 1.89 |
| Shanghai Jiao Tong University | China | 25 | 168 | 6.72 | 7 | 8.0% | 11.50 | 6.30 |
| Tianjin University | China | 22 | 158 | 7.18 | 6 | 4.5% | 0.00 | 7.52 |
| Tongji University | China | 21 | 120 | 5.71 | 6 | 14.3% | 14.33 | 4.28 |
| Newcastle University | United Kingdom | 19 | 714 | 37.58 | 10 | 31.6% | 20.33 | 45.54 |

(C): country (A): number of articles; (TC) number of citations (TC/A): average number of citations per article; (H-index): Hirsch index for this research area (IC%): percentage of articles written through international cooperation; (IC): number of citations per article written through international cooperation; (NIC): number of citations per article written without any cooperation.

It is worth noting that all of the institutions considered have international cooperation rates that are less than 50%, indicating that the institutions have produced very few articles through international cooperation, corroborating the analysis shown in Figure 4 that indicates that most authors are publishing with others from the same country and even from the same institution.

The total citations per article (TC/A) with international cooperation (IC) was higher than the total citations per article (TC/A) without international cooperation (NIC). It is extremely important to create international cooperation networks for research, as they could help research have a greater impact by analyzing citations of articles written through international cooperation and without this cooperation. The disparity in the quantity of citations is notable. In some cases, articles that are written through a process of international cooperation are cited more than three times as often as those written without cooperation, as in the case of Beijing Jiaotong University, Harbin Engineering University, and Tongji University, which had more than three citations in articles that were published in an international partnership.

The countries that were most prolific in publishing work on rail systems between 2002 and 2021 are presented in Table 6. China and the United States emerge as the most prolific and important countries, since, together, they represent almost 80% of all the work published on this subject, with 571 articles and 4145 citations and 361 articles and 7359 citations, respectively. The United Kingdom contributed with 168 articles and 3799 citations, which leaves it in third place.

**Table 6.** Most prolific countries for the publication of articles on rail systems between 2002 and 2021.

| Country | A | TC | TC/A | H Index | R | | | |
| | | | | | 2002–2006 | 2007–2011 | 2012–2016 | 2017–2021 |
|---|---|---|---|---|---|---|---|---|
| China | 571 | 4145 | 7.26 | 30 | (1) 46 | (1) 88 | (1) 158 | (1) 279 |
| United States | 361 | 7359 | 20.39 | 46 | (5) 10 | (2) 87 | (2) 119 | (2) 111 |
| United Kingdom | 168 | 3799 | 22.61 | 31 | (2) 44 | (3) 35 | (3) 53 | (3) 62 |
| Germany | 91 | 591 | 6.49 | 16 | (4) 18 | (4) 21 | (8) 24 | (9) 15 |
| Italy | 85 | 1479 | 17.40 | 22 | (3) 31 | (5) 13 | (7) 25 | (4) 37 |
| Australia | 70 | 1135 | 16.21 | 21 | (7) 9 | (7) 8 | (4) 31 | (5) 26 |
| Spain | 62 | 1075 | 17.34 | 18 | (8) 5 | (8) 8 | (5) 26 | (6) 23 |
| Canada | 55 | 881 | 16.02 | 18 | (9) 5 | (9) 8 | (9) 16 | (7) 21 |
| South Korea | 51 | 420 | 8.24 | 12 | (6) 10 | (10) 6 | (6) 26 | (10) 15 |
| Taiwan | 42 | 701 | 16.69 | 14 | (10) 4 | (6) 11 | (10) 12 | (8) 16 |

(A): number of articles (R): ranking from the number of articles in 5 years; (TC): number of citations provided; (TC/A): number of citations per article; (H-Index): Hirsch index rating the research area.

China, the United States, and the United Kingdom are the only countries that have contributed more than 100 articles each on rail systems. However, it is important to note that this is a subject that generates widespread interest, and contributions to the field of study have come from many countries from different continents.

In terms of the evolution of the study area, China produced the greatest number of scientific research articles during the final five-year period, which illustrates its obvious ongoing development of scientific research on this issue.

The United Kingdom took second place in the first five-year period, with 44 publications, but this country is losing production in comparison to the United States, which took second place in the next five years and remained there until the end of 2020.

It is worth noting that, in the final five-year period, almost all of the countries that were considered increased their research output on this subject; however Spain, Germany, and the United States are exceptions, as their output decreased. China, which is at the top of the production ranking, doubled its publication rate in the final five-year period.

Table 7 shows the effective international networks and the metrics evolution of the countries that produced the most research on the study area during the period under consideration.

**Table 7.** Cooperation between different countries for research on rail systems between 2002 and 2021.

| Country | NC | Main collaborators | IC (%) | TC/A | |
| --- | --- | --- | --- | --- | --- |
| | | | | IC | NIC |
| China | 25 | United States, United Kingdom, Hong Kong, Australia, Canada | 16.8% | 15.74 | 5.55 |
| United States | 34 | United States, China, United Kingdom, Australia, Canada | 27.1% | 23.37 | 19.27 |
| United Kingdom | 27 | China, United States, Australia, Italy, Germany | 38.1% | 28.59 | 18.93 |
| Germany | 14 | United Kingdom, Germany, Spain, United States, Canada | 22.0% | 14.15 | 4.34 |
| Italy | 21 | Austria, Italy, Spain, United Kingdom, Netherlands | 30.6% | 13.50 | 19.12 |
| Australia | 15 | United Kingdom, United States, China, Canada, Iran | 38.6% | 20.41 | 13.58 |
| Spain | 11 | Netherlands, United Kingdom, United States, Italy, France | 33.9% | 24.19 | 13.83 |
| Canada | 18 | China, United States, Australia, Hong Kong, Italy | 41.8% | 16.04 | 16.00 |
| South Korea | 8 | United States, India, Thailand, Malaysia, Singapore | 27.5% | 15.29 | 5.57 |
| Taiwan | 14 | United States, Australia, Brazil, Hong Kong, Canada | 23.8% | 22.00 | 15.03 |

(NC): number of countries cooperating; (CI): percentage of countries involved in cooperatively created articles; (TC/P): number of citations per article; (IC): international cooperation; (NIC): non-international cooperation.

Canada took part in more cooperative work than any other country, with a percentage of 41.8%, followed by Australia and the United Kingdom, with 38.6% and 38.1%, respectively. However, China, which is the country with the highest number of publications, has the lowest number of international partnerships, ranking last at 16.8%.

Despite having a large number of international collaborators, all countries have a rate of international cooperation that is less than 50%. This means that more than half of their publications are written by authors from the same country. According to a citation analysis of the articles, the value of citation increases significantly when there is international cooperation (IC × NIC).

Figure 5 illustrates a map of collaboration among the major countries based on co-authorship, with at least six interactions. It is represented by nine clusters of different colors; the colors represent international work groups, and the size of the circle varies depending on the number of articles.

The red and green clusters are the largest clusters. It is comprised of seven countries, which are led by India, South Korea, Germany, and Russia, respectively. This is followed by the blue, yellow, and purple clusters, which are each comprised of five countries, and they are led by Australia, Spain, and the United Kingdom, respectively. The light blue cluster, which is comprised of four countries, is led by Japan, and, finally, the orange, brown, and lilac clusters, which are each comprised of only two countries, are led by China, Canada, and the United States, respectively.

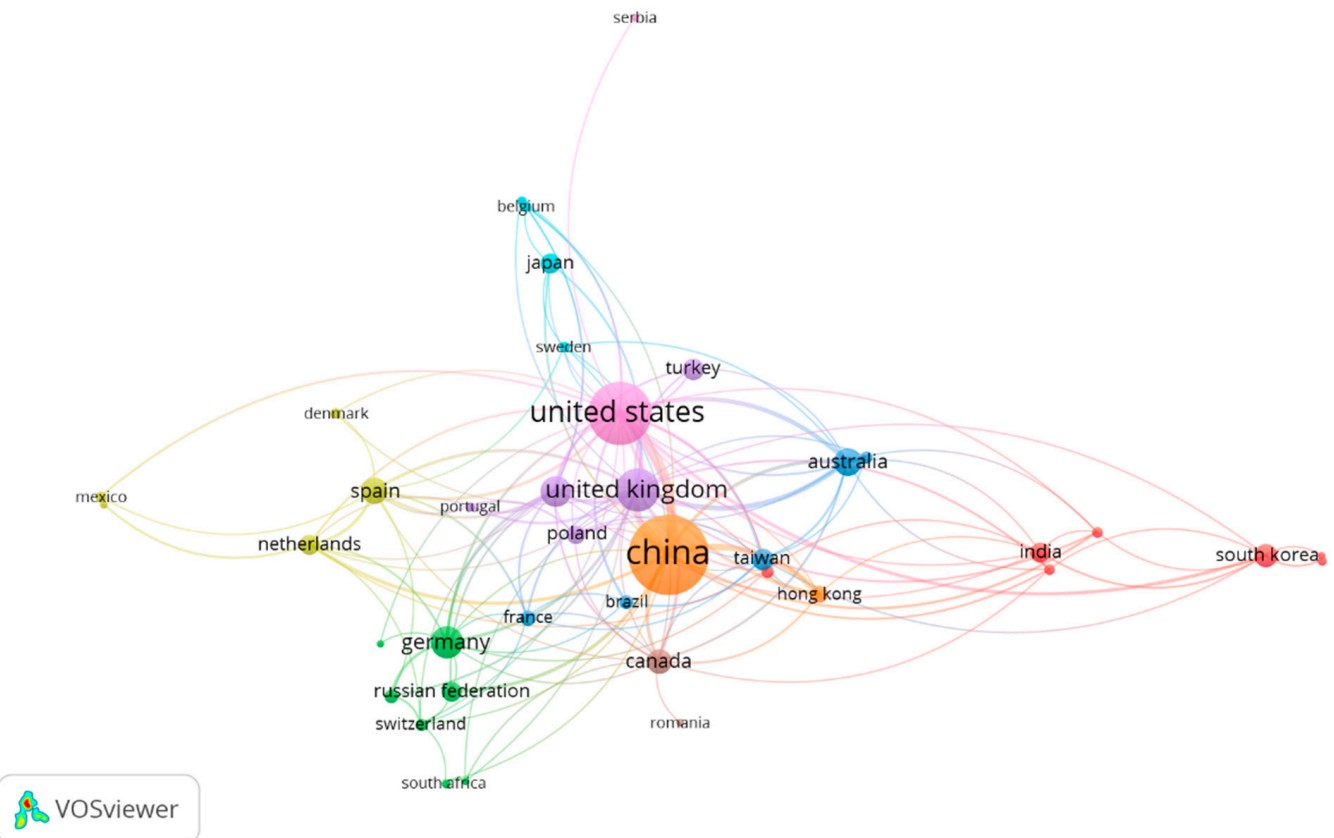

**Figure 5.** International cooperation, based on international co-authorship.

*3.4. Main Research Areas on Rail Systems (Q4)*

In order to study rail systems, an analysis of the keywords was carried out. From the 1914 articles on railway systems from the period between 2002 and 2021 that were analyzed, a total of 7711 keywords were obtained.

Keywords represent the content of a document. Keyword analysis is a useful method for knowledge mining and gaining insight into the structure of knowledge and research trends [40].

Table 8 lists the 20 keywords that were most relevant to the authors of this research, as they allow the interests that were generated in the line of research to be disaggregated. To facilitate in the analysis of key words, six thematic groups were applied.

The "maintenance" group is a thematic group that was studied in depth, with the keywords "Light Rail Transit" found in around 9.5% of publications. The evolution of this research increased over the years, and the largest number of articles published on this subject were found in the 2017–2021 period.

Urban transport development, train maintenance, and combustion are the areas that were studied the most and were written about the most. From these three sections, the development of urban transport is the area on which the most research has been published.

According to the dimensioning of the group "Energy Efficiency and Greenhouse Gases", the environment continues to be a neglected area by researchers. Sustainability, environmental effects, and climate change are topics that are rarely addressed in railway research. However, the last five years have seen a significant rise in the study of this field.

**Table 8.** Keywords for rail systems.

| Group | Keyword | 2002–2021 | | 2002–2006 | | 2007–2011 | | 2012–2016 | | 2017–2021 | |
|---|---|---|---|---|---|---|---|---|---|---|---|
| | | A | % | A | % | A | % | A | % | A | % |
| Sustainability and Rail Systems | Urban Development | 37 | 1.9% | 3 | 1.2% | 7 | 1.8% | 11 | 1.9% | 16 | 2.3% |
| | Optimization | 68 | 3.5% | 7 | 2.8% | 13 | 3.4% | 21 | 3.6% | 27 | 3.9% |
| | Light Rail | 118 | 6.2% | 3 | 1.2% | 19 | 4.9% | 26 | 4.5% | 70 | 10.0% |
| | Environmental Impact | 22 | 1.1% | 4 | 1.6% | 5 | 1.3% | 8 | 1.4% | 5 | 0.7% |
| Railway Mechanics | Common Rail System | 92 | 4.8% | 9 | 3.6% | 17 | 4.4% | 27 | 4.6% | 39 | 5.6% |
| | Wheel-rail Systems | 82 | 4.3% | 9 | 3.6% | 10 | 2.6% | 18 | 3.1% | 45 | 6.4% |
| | Finite Element Method | 63 | 3.3% | 8 | 3.2% | 10 | 2.6% | 18 | 3.1% | 27 | 3.9% |
| | High-speed Train | 78 | 4.1% | 4 | 1.6% | 10 | 2.6% | 9 | 1.5% | 20 | 2.9% |
| Urban Train Development | Accessibility | 38 | 2.0% | 1 | 0.4% | 7 | 1.8% | 12 | 2.1% | 18 | 2.6% |
| | Public Transport | 112 | 1.8% | 12 | 4.8% | 32 | 8.3% | 22 | 3.8% | 46 | 6.6% |
| | Rail Systems | 180 | 9.4% | 9 | 3.6% | 82 | 21.2% | 40 | 6.9% | 49 | 7.0% |
| | Sustainability | 38 | 2.0% | 2 | 0.8% | 6 | 1.6% | 11 | 1.9% | 19 | 2.7% |
| Maintenance | Urban Transport | 132 | 6.9% | 16 | 6.5% | 27 | 7.0% | 42 | 7.2% | 47 | 6.7% |
| | Light Rail Transit | 183 | 9.5% | 19 | 7.7% | 30 | 7.8% | 64 | 11.0% | 70 | 10.0% |
| | Transportation Infrastructure | 51 | 2.7% | 4 | 1.6% | 6 | 1.6% | 20 | 3.4% | 21 | 3.0% |
| | Vibrations (mechanical) | 92 | 4.8% | 9 | 3.6% | 15 | 3.9% | 16 | 2.7% | 52 | 7.4% |
| Energy Efficiency and Greenhouse Gases | Energy Use | 34 | 1.8% | 3 | 1.2% | 5 | 1.3% | 10 | 1.7% | 16 | 2.3% |
| | Greenhouse Gases | 20 | 1.0% | 0 | 0.0% | 5 | 1.3% | 7 | 1.2% | 8 | 1.1% |
| | High-speed Rail | 31 | 1.6% | 1 | 0.4% | 4 | 1.0% | 10 | 1.7% | 16 | 2.3% |
| | Emission Control | 20 | 1.0% | 0 | 0.0% | 4 | 1.0% | 5 | 0.9% | 11 | 1.6% |
| Combustion | Diesel Engines | 245 | 12.8% | 45 | 18.1% | 56 | 14.5% | 65 | 11.1% | 79 | 11.3% |
| | High-pressure Effects | 24 | 1.3% | 4 | 1.6% | 3 | 0.8% | 7 | 1.2% | 10 | 1.4% |
| | High-pressure Common Rail System | 97 | 5.1% | 6 | 2.4% | 22 | 5.7% | 32 | 5.5% | 37 | 5.3% |
| | Fuel Injection | 87 | 4.5% | 21 | 8.5% | 12 | 3.1% | 23 | 3.9% | 31 | 4.4% |
| | Total number of articles: | 1.918 | | 248 | | 387 | | 583 | | 700 | |

(A): the number of articles with the keywords match.

Figure 6 illustrates the rail system keywords that were used in the 20-year period, which were considered based on the co-occurrence method. Keywords with at least five interactions were chosen. Keywords identify the major points of a study and describe research subjects in a certain field. The close links between terms are represented by their co-occurrence.

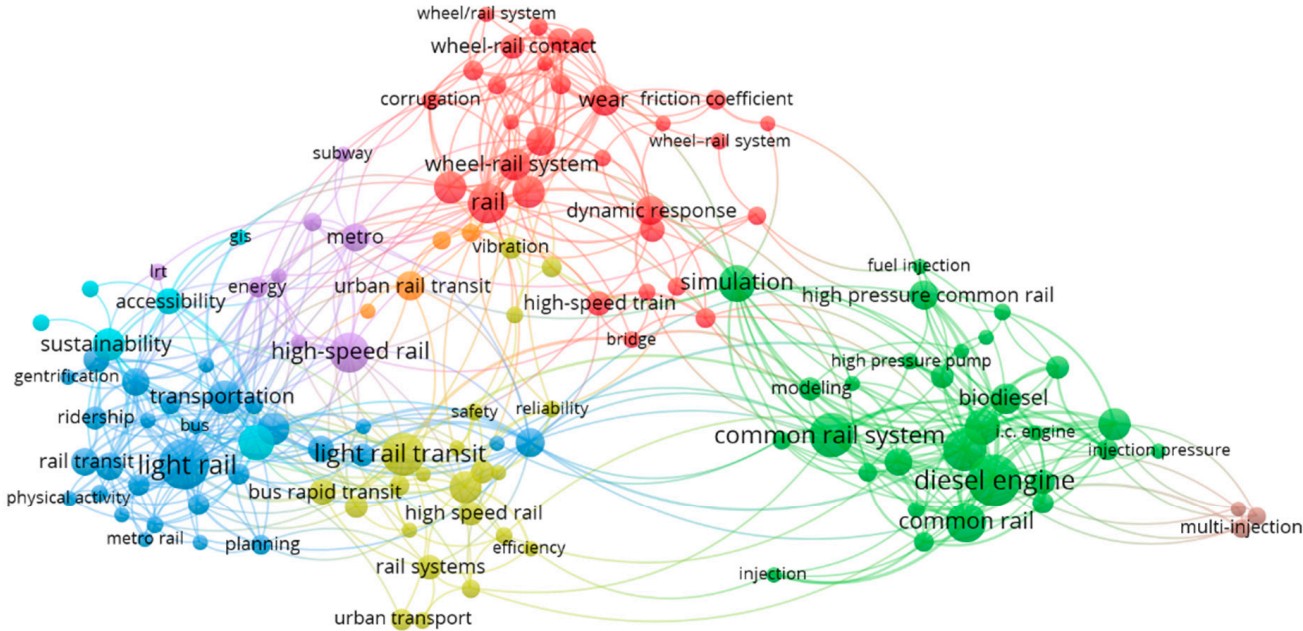

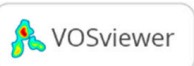

**Figure 6.** Keywords in co-occurrence-based rail system publications.

The exploration of important research subjects and emerging research trends using the co-occurrence of these terms is a very successful research strategy.(Rajeswari, S. et al., 2021). By using VOSviewer, a visual representation of networks based on distance was generated, where the distance between two nodes denotes the degree of their proximity. By using this approach. keywords in our search domain could be identified [41].

The red cluster, which has the highest number of keywords (30), refers to the study of the mechanical engineering trains, the main objective of which is to research improvement in the mechanical movement of the railway system.

Many studies focus on steps such as computer simulation or the development of new technologies to improve the process of wave patterns, friction in the contact zone, vibration, and dynamic component distribution patterns during the train's movement over the railway to obtain more speed, improved performance, safety, and the performance of the locomotor [42–54].

The green cluster, with 29 keywords, refers to the study of engines, energy consumption, new sustainable types of fuel [55], new types of combustion systems [56,57], biodiesel [58], natural gas [59], and liquefied petroleum gas [60]. Many articles presented research focused on internal combustion engines, which is a heat engine in which the combustion of a fuel with an oxidant takes place in a combustion chamber that is an integral part of the working fluid flow circuit. Basically, in an engine, chemical energy is transformed into thermal energy through fuel combustion, and this thermal energy is transformed into mechanical energy that is used to perform train movement [61]. These studies focus on new engine models, such as multi-fuel, biofuel, and water-powered engines. Research can also be found that aims to improve diesel engine pressure by using simulation and modeling [61–65].

The dark blue, with 29 keywords, addresses research in the areas of urban passenger trains, urban planning [66,67], light vehicle performance [68], new technologies to improve the energy efficiency of light vehicles [69], the relationship between the public and trains [70,71], and investment in public urban transport policies [72,73].

The yellow cluster, with 22 keywords, represents the areas of preventive and predictive maintenance to ensure passenger safety [74], rail system efficiency as a calculation of nodal capacity utilization rates, capacity, railway station performance [75–77], the efficiency of the urban transport system [78], and transport regulation in different countries.

The purple cluster, with only eight keywords, addresses the sustainable development of rail systems, such as reducing greenhouse gas emissions [79], life cycle analysis [80,81], the development of new types of sustainable energy, and the environmental impact of rail systems, which mainly focuses on air pollution [82,83]. The environmental impact of railway systems has not been studied in depth, and, therefore, it is an area that presents new research opportunities.

The light blue cluster has only six keywords and investigates the sustainability index [2,84], accessibility of urban transport [85], and the lack of urban train accessibility in different cities [86–88].

The orange cluster of keywords addresses the concept of improving urban trains, urban mobility, and urban planning as options for accessing and exiting urban rail transport [89].

Finally, the smallest cluster in brown, with only three keywords, interacts with the green cluster and refers to direct fuel injection systems, multi-material injection modeling, and modeling of restrictor valve performance and engine performance [90].

## 4. Conclusions

This work analyzed the evolution of research articles on railway systems with a 20-year global retrospective that covered the period between 2002 and 2021. Even though rail systems have been developed for centuries, the scientific production in this topic has started to grow only in the last 20 decades. This growth is followed by an increase in the number of journals, institutions, and authors that publish on the subject, which demonstrates a recent increase in the scientific community's interest in the theme. Most of the scientific productions come from China; however, a lot of research interaction was observed between other countries.

The bibliometric study found 1918 articles published with the keywords "Rail System" between 2002 and 2021. Throughout the study period analyzed, Engineering was the category that was studied the most in the rail system field, followed by Social Sciences and Environmental Sciences, Computer Science, and Material Science.

When analyzing authors, the researcher Zhang, Y.T. was found to be the most important one in the whole period, with 15 articles published. It is interesting to note that, among the 10 authors who published the most, 9 are from Chinese institutions.

Mulley, C., Chen, G.X., and Pagliara are strong authors who contribute to international cooperation in transportation systems among the top ten most productive authors. However, it should be highlighted that interactions between writers from the same nation are far more common. This is a poor rail system search index because the quick expansion of the research field and the transmission of knowledge may be facilitated by collaboration with other international authors.

When analyzing scientific journals, *Transportation Research Record* was the most productive during the period between 2002 and 2021, but the journal *Fuel* had the highest H-index rating. *Transportation Research Part A: Policy And Practice* had the highest average number of citations per articles (49.03), even though it was in fourth place in terms of number of publications.

The fact that at least one article was published in at least 55% of the journals consistently throughout the course of five years indicates that railway systems were thoroughly researched at this time.

The Chinese institutions predominated in the publication production, with nine institutions from the ten most prolific institutions. All of the institutions analyzed had results of under 50% in terms of cooperation, which is an important matter to be considered to enhance research quality.

The main areas of research on railway systems are mechanical engineering, engines, fuels, maintenance, and the transport planning of urban trains. There has been a significant surge in research productivity on the environmental impact and sustainability of rail systems; however, this is very low in comparison to other topics.

Researchers can use this study to emphasize various concepts and relationships between them, opening up new research possibilities. In this instance, the article identifies a number of study trends. The first is the expanding significance of the areas' attention to actions such as computer modeling or the creation of new technologies to enhance wave patterns, friction in the contact zone, and vibration. The second is the small number of studies that have been conducted for creating approaches for the environment and sustainability in the railroad. To fill in the gaps between what might be a potential subject for future work, more studies in this area are required.

Urban planning decision makers will also benefit from this research because its findings will enable them to identify the most efficient systems and technologies, as well as the approaches for more accurate railway evaluation.

## 5. Limitations and Future Work

One drawback of the strategy is that some pertinent research may have been missed by the Scopus database. Additionally, informal routes derived from the data search—such as oral contact between scientists, internal reports between research institutes, or other channels—that were excluded from the study can be equally significant.

Conducting a more thorough analysis and providing a more thorough discussion of the results that have been introduced in the preceding sections would be an interesting direction for future work. This could be carried out with cutting edge bibliometric tools, and any appropriate statistical test can be provided to make the experimental analysis more discernible.

**Author Contributions:** Conceptualization, D.D.F.-S. and A.F.R.-R.; methodology, D.D.F.-S. and A.F.R.-R.; software, D.D.F.-S.; validation, D.D.F.-S., A.F.R.-R., J.D.G. and S.A.E. formal analysis, D.D.F.-S.; investigation, D.D.F.-S.; resources, D.D.F.-S.; data curation, D.D.F.-S.; writing—original draft preparation, D.D.F.-S.; writing—review and editing, A.F.R.-R., J.D.G. and S.A.E.; visualization, A.F.R.-R., J.D.G. and S.A.E.; supervision, A.F.R.-R., J.D.G. and S.A.E.; project administration, A.F.R.-R., J.D.G. and S.A.E. All authors have read and agreed to the published version of the manuscript.

**Funding:** This research received no external funding.

**Institutional Review Board Statement:** Not applicable.

**Informed Consent Statement:** Not applicable.

**Data Availability Statement:** The data presented in this study are available in this article.

**Conflicts of Interest:** The authors declare no conflict of interest.

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
