# Peer review of "A Bibliometric Analysis of the Trends and Characteristics of Railway Research"

_sustainability, doi:10.3390/su142113956_

Round 1

Reviewer 1 Report

 "A bibliometric analysis on trends and characteristics of railway Research" has been carefully reviewed and several issues have been found.

This paper has studied a retrospective bibliometric analysis of the railway sector covering the 20-year period between 2002 and 2021 was carried out to better understand the characteristics of the railway research. The Scopus database contained 1,918 articles published with the keywords “Rail System”.

VOSviewer software was used to create network maps from each of the variables studied. The results showed a huge increase in the number of publications over this period, notably the work written by Zhang, Y.T., who was found to be the most productive author. Engineering was found to be the most studied subject area of knowledge; Transportation Research Record was the journal with the highest number of publications; China was revealed to be the leading country in this research field, and Southwest Jiaotong University was the leading institution in this topic. Finally, there was a lack of research on the environmental impact and sustainability of railway systems, an area which could be opened up for future study

This paper might be of some interest, but the content is not novel at all. It describes a procedure to find out the bibliometric analysis of railway Research. 

However, from my point of view, the authors have not highlighted the innovation of the manuscript enough.

Author Response

Dear Reviewer

Thank you so much for reading my work and helping scholars all over the world publish their articles in a much more organized and scientific way by taking the time out of your day to do so.
Below are some remarks from the reviewer's suggestion

Point: This paper might be of some interest, but the content is not novel at all. It describes a procedure to find out the bibliometric analysis of railway Research.

Response: This work is novel because there hasn't been another bibliometric study with the keyword "rail systems" (in the past 20 years). It is vital to note that this work will be useful to other railroad scholars because it will outline the current research in the field and identify any gaps that exist.

The article even resulted in the main areas of research on railway systems are mechanical engineering, engines, fuels, maintenance, and transport planning of urban trains. There has been a significant surge in research productivity on the environmental impact and sustainability of rail systems, however very low in comparison to other topics. More study is needed to bridge the gap between what might be a promising topic for future work.

The information described in this article may redirect investigations in the railway area.

Reviewer 2 Report

1. To enrich the literature study include a relevant research article as provided below

https://www.hindawi.com/journals/jat/2022/7685375/https://www.hindawi.com/journals/jat/2022/7685375/

https://sciendo.com/article/10.2478/ttj-2018-0010

2. Corresponding to FIGURE 3, the data can be represented in pie charts for a better understanding

3. Any suitable statistical test can be provided to make the experimental analysis more acceptable

4. The conclusion should be more result oriented

5. Future scope of the study should be appended in a new paragraph under the Conclusion section.

Author Response

Dear reviewer

First, I want to thank the reviewer for the suggestions for the article.
Thank you so much for reading my work and helping scholars all over the world publish their articles in a much more organized and scientific way by taking the time out of your day to do so.

The recommendations were adopted and used in the article, but the following details:

Point 1: To enrich the literature study include a relevant research article as provided below (https://www.hindawi.com/journals/jat/2022/7685375/https://www.hindawi.com/journals/jat/2022/7685375/; https://sciendo.com/article/10.2478/ttj-2018-0010)

Response 1: The two kinds of literature were included, as suggested by the reviewer.

Point 2:  Corresponding to FIGURE 3, the data can be represented in pie charts for a better understanding.

Response 2: The pie chart was made, but it did not represent the 3 information (Subject área, year, total of articles), therefore, the graph has not been modified.

Point 3: Any suitable statistical test can be provided to make the experimental analysis more acceptable

Response 3: Include any suitable statistical test to provide to make the experimental analysis more acceptable was a great idea, but it will be used in future studies and I include it in the Future scope section.

Point 4:  The conclusion should be more result oriented

Response 4: It was included in the conclusion as more “result-oriented”, as the reviewer requested.

Point 5:  Future scope of the study should be appended in a new paragraph under the Conclusion section.

Response 5: Future scope of the study was appended in a new paragraph under the Conclusion section, as the reviewer requested.

Reviewer 3 Report

Thank you for the opportunity to read the paper. It is an interesting toping and I consider it fits to the journal. very well organized, in conceptual and methodological terms, and presents very relevant results

Within the limits of the possibilities, the following mapping can be done:

  • Mapping of articles co-citations
  • Mapping of journal co-citations
  • Mapping of institutions’ co-citations

 The bibliography is up to date, but a possible source can be:

Radu, V.; Radu, F.; Tabirca, A.I.; Saplacan, S.I.; Lile, R. Bibliometric Analysis of Fuzzy Logic Research in International Scientific Databases. Int. J. Comput. Commun. Control 2021, 16, 1–20.

Author Response

Dear reviewer 

First, I want to thank the reviewer for the suggestions for the article.
Thank you so much for reading my work and helping scholars all over the world publish their articles in a much more organized and scientific way by taking the time out of your day to do so.

The recommendations were adopted and used in the article, but the following details:

Point 1: Within the limits of the possibilities, the following mapping can be done: Mapping of articles co-citations; Mapping of journal co-citations; Mapping of institutions’ co-citations.

Response 1: This article's goal was to discover networks of international cooperation based on co-citations; as neither the journals nor the articles contributed to this goal, for this reason, it was not incorporated in the article. However, crucial data that assisted in achieving the study's goal may be found in the tables (TC/A) average number of citations per article; (H-index) Hirsch index for this research area (IC%) percentage of articles written through international cooperation; (IC) number of citations per article written through international cooperation; (NIC) number of citations per article written without any cooperation) and the maps (International cooperation, based on international co-authorship, Co-authoring network of authors, Keywords in co-occurrence-based rail system publications).

Point 2: The bibliography is up to date, but a possible source can be:

Response 2: The literature suggested “Radu, V.; Radu, F.; Tabirca, A.I.; Saplacan, S.I.; Lile, R. Bibliometric Analysis of Fuzzy Logic Research in International Scientific Databases. Int. J. Comput. Commun. Control 2021, 16, 1–20” were included in the article.

Round 2

Reviewer 1 Report

The author has done good work and covered many accept of trends and characteristics of railway research. Hence, I accept